# Robust IMU-Based Mitigation of Human Body Shadowing in UWB Indoor Positioning

**DOI:** 10.3390/s23198289

**Published:** 2023-10-07

**Authors:** Cedric De Cock, Emmeric Tanghe, Wout Joseph, David Plets

**Affiliations:** Department of Information Technology, IMEC-WAVES/Ghent University, Technologiepark-Zwijnaarde 126, 9052 Gent, Belgium; emmeric.tanghe@ugent.be (E.T.); wout.joseph@ugent.be (W.J.); david.plets@ugent.be (D.P.)

**Keywords:** indoor localization, UWB, IMU, human body shadowing, particle filter, Gaussian mixture model

## Abstract

Ultra-wideband (UWB) indoor positioning systems have the potential to achieve sub-decimeter-level accuracy. However, the ranging performance degrades significantly under non-line-of-sight (NLoS) conditions. The detection and mitigation of NLoS conditions is a complex problem and has been the subject of many works over the past decades. When localizing pedestrians, human body shadowing (HBS) is a particular and specific cause of NLoS. In this paper, we present an HBS mitigation strategy based on the orientation of the body and tag relative to the UWB anchors. Our HBS mitigation strategy involves a robust range error model that interacts with a tracking algorithm. The model consists of a bank of Gaussian Mixture Models (GMMs), from which an appropriate GMM is selected based on the relative body–tag–anchor orientation. The relative orientation is estimated by means of an inertial measurement unit (IMU) attached to the tag and a candidate position provided by the tracking algorithm. The selected GMM is used as a likelihood function for the tracking algorithm to improve localization accuracy. Our proposed approach was realized for two tracking algorithms. We validated the implemented algorithms on dynamic UWB ranging measurements, which were performed in an industrial lab environment. The proposed algorithms outperform other state-of-the-art algorithms, achieving a 37% reduction of the p75 error.

## 1. Introduction

Indoor Positioning System (IPSs) track people or objects in GNSS-denied environments, i.e., inside buildings, ships, multi-level parking lots, etc. Indoor localization knows many applications, e.g., automatic inventarization with drones [1] and transportation with AGVs [2], protecting factory workers from collisions [3], tracking staff, patients and equipment in hospitals [4], etc. The ubiquity of smartphones in our daily lives, as well as the presence of Wi-Fi access point (APs) in most public and office buildings, has created the opportunity for IPSs to use available infrastructure. As such, IPSs are being developed, with which people can use their own smartphones to navigate through public buildings, e.g., museums [5]. The presence or absence of traffic between smartphones and Wi-Fi APs or Bluetooth Low Energy (BLE) beacons, along with other sensor data such as indoor CO_2_ levels and illuminance, can be used for room occupancy detection [6]. Positional data can be used directly, e.g., by emergency responders to find people in distress [7], but also for purposes other than navigation or guiding, allowing for endless amounts of applications. Room occupancy detection can be used for the automatic control of smart plugs [8] or actuation of heating, ventilation, and air conditioning [9] to reduce energy consumption. Another example is the automatic activity recognition and performance analysis of athletes using UWB positioning, with which a coach could provide personalized feedback for the whole team [10].

In most cases, a mobile node (tag) is localized relative to a set of fixed nodes (anchors) using wireless technologies, e.g., Wi-Fi [11], BLE [12], Ultra-Wideband (UWB) [13], Visible Light Positioning (VLP) [14], etc. Unlike GNSS in outdoor scenarios, there is no standardized solution that can be used for all applications. Instead, the technology is chosen based on the required accuracy, hardware, installation, and maintenance cost, as well as the complexity of the localization problem. As mentioned, using BLE beacons and Wi-Fi APs as proximity-based sensors allow for the design of cheap and easily deployable IPSs with room-level accuracy [8,9]. Depending on the node density, BLE and Wi-Fi fingerprinting can offer an accuracy of 2–3 m but require extensive measurement campaigns or knowledge of the building materials for empirical [15] or model-based [16] fingerprinting, respectively. In time-of-flight (ToF) ranging, the travel time of a signal is measured to estimate the distance between a tag and anchor [13]. The arrival time of a UWB signal can be accurately measured due to its narrow pulses, allowing for centimeter-level ranging accuracy. These narrow pulses also make UWB ranging immune to multipath fading, unlike Wi-Fi and BLE ranging. Although the arrival of Commercial Off-the Shelf (COTS) UWB transceivers [17] has made UWB IPSs more affordable, they do require a dedicated tag and node infrastructure, which increases the cost and deployment effort. Furthermore, VLP is an emerging localization technology that offers similar accuracy to UWB at a lower cost [18]; however, receiver tilt as well as blockage of the receiver can degrade its performance, which makes pedestrian tracking a complex problem for VLP-based IPS. Because this work is aimed at the accurate tracking of pedestrians in the industrial and sports contexts, we choose to use UWB ranging with the tag attached to the torso, i.e., on-body UWB pedestrian localization.

While UWB can achieve high ranging accuracy in line-of-sight (LoS) conditions, there is still ongoing research in mitigating the effects of non-line-of-sight (NLoS) conditions. Under these conditions, the direct path between the tag and an anchor is (partly) obstructed by a wall, object, or the human body. Depending on the obstruction, it is possible that the received power along the direct path is too low, and through reflection or diffraction, an indirect path component of the signal is detected instead [19]. Because the indirect path travels a larger distance than the direct unobstructed path, the signal is detected with a delay, causing a positive bias on the range estimation [13].

Human Body Shadowing (HBS) is a specific but important case of NLoS in pedestrian tracking, in which a pedestrian carries the UWB tag and obstructs the LoS path with their body. Preliminary works have investigated the impact of HBS in UWB ranging and have concluded that the UWB range error distribution changes with the orientation of the body and tag relative to the anchor [19,20]. Aside from the orientation, the error distribution is shown to depend on the position of tag on the body [21] and the distance of the tag from the body [22]. Based on the tag position on the body, several orientation-specific error distributions were proposed [23].

Given these findings, it is clear that knowledge of the relative orientation between the human body and the UWB tag–anchor provides an opportunity to detect HBS-induced range errors for on-body UWB positioning. However, there is only limited research that uses this orientation to reduce the impact of HBS effects on on-body UWB-based pedestrian tracking. In fact, while many methods have been proposed to (detect and) mitigate NLoS conditions, the differentiation between different types of NLoS is not often performed. More specifically, recent research is primarily focused on using deep-learning techniques, e.g., Convolutional Neural Network (CNNs) [24], to identify general NLoS conditions in the Channel Impulse Response (CIR). There is also a growing interest to fuse measurements of an Inertial Measurement Unit (IMU), which can provide absolute orientation, with UWB measurements in order to improve UWB pedestrian tracking [25,26]. However, these methods employ the IMU to predict positions using Pedestrian Dead Reckoning (PDR) [26] or the integration of the inertial data [25], independent of any type of UWB NLoS condition. The few works that do perform orientation-aware UWB HBS mitigation have clear limitations [27,28], which are discussed in Section 2. Therefore, our work addresses this general lack of research on this topic and improves on the limitations of the few existing related works.

In this work, the heading provided by an IMU’s Attitude and Heading Reference System (AHRS) and the estimated position of a UWB tracking algorithm are combined in order to estimate the relative human body orientation. After estimating this orientation, the impact of HBS-induced range errors on the localization system is mitigated by a newly proposed robust Gaussian Mixture Model (GMM)-based, orientation-aware range error model. This two-part HBS mitigation strategy is then integrated into a Particle Filter (PF) and Gaussian Sum Filter (GSF) algorithm. The performance of the resulting UWB IPSs are evaluated with dynamic measurements in an industrial lab environment using a Decawave transceiver and an Adafruit BNO055 IMU. The performance is benchmarked against state-of-the-art systems, and the impact of each of the two parts of our mitigation strategy is investigated by comparing it with alternative solutions. Furthermore, the runtime of each algorithm is analyzed, as well as the impact of anchor selection and smoothing, in addition to the proposed mitigation strategy.

In summary, the following contributions have been realized:Synergism of on-body IMU and UWB Two-Way Ranging (TWR) measurements for the accurate estimation of body–tag–anchor orientation and mitigation of human body shadowing effects.A robust GMM-based, orientation-aware range error model for the mitigation of human body shadowing effects in on-body UWB-TWR pedestrian tracking.Integration of the GMM-based range error model with a Gaussian Mixture Filter, which provides higher localization accuracy than the state-of-the-art methods while reducing the computation cost by an order of magnitude.Our proposed algorithms have been evaluated and benchmarked against a state-of-the-art algorithm [27] based on measurements with mm-level motion capture (mocap) ground truth. An accurate ground truth was not available in [27].

The remaining part of the paper starts with a discussion of related works (Section 2), followed by a description of the experiment setup and proposed HBS mitigation method (Section 3). The performance of the implemented algorithms is discussed in Section 4, after which a conclusion is formulated in Section 5.

## 2. Related Work

An overview of the works investigating the effect of the human body on UWB ranging is provided in Section 2.1. Section 2.2 summarizes general NLoS detection and mitigation approaches in the UWB ranging context; it also discusses the limited amount of works that are specifically focused on HBS detection and mitigation, as well as the motivation for our proposed work. Furthermore, it is necessary to define the mentioned body–tag–anchor orientation itself.

As in our previous work [29], the relative body orientation is defined in Figure 1 as the angle ϕ∈[0°, 180°] between vector $\vec{PT}$ and vector $\vec{TA}$. $\vec{PT}$ and $\vec{TA}$ are 2D vectors in the horizontal plane, of which the points *P*, *T*, and *A* represent the 2D locations of the person’s midpoint, tag antenna, and anchor antenna, respectively. It is assumed that the tag is placed in such way that both the radiation pattern of the tag antenna and the influence of the body are symmetrical with respect to $\vec{PT}$. Other elements of Figure 1 are discussed in Section 3.

### 2.1. Effects of Human Body Shadowing on UWB Ranging

No significant shadowing effects on UWB ranging have been observed for $\phi <67.5^{\circ }$ with a chest-mounted tag [22] and $\phi <90^{\circ }$ with the tag being held at chest height a few cm in front of the body [30]. The average range error increases from 10 cm to 20 cm for $67.5^{\circ }<\phi <112^{\circ }$ in [22], while [30] reports a similar increase for $90^{\circ }<\phi <155^{\circ }$. While the average range error increases, the standard deviation of the range error is unchanged for these intervals, and the error Probability Density Function (PDF) still resembles a Gaussian function [30]. For $\phi \to 180^{\circ }$, the average error increases to 60 cm for $\phi >112^{\circ }$ in [22], with [30] again having similar results for $\phi >150^{\circ }$. Range errors of up to 3 m were measured with a chest-mounted tag for true distances below 3 m in [21].

This behavior is explained by the findings of studies on the effects of HBS on the received power of UWB signals [19,22]. As with ranging, no significant effects on the received power are observed when the body does not fully obstruct the LoS path [22]. When the latter occurs, the signal reaches the receiver by diffracting around the body (i.e., a creeping wave) or by reflecting off another surface [19]. As the creeping waves are more attenuated for higher ϕ angles, reflections become more likely to be recognized as the first path by the receiver. In fact, the attenuation of creeping waves for $\phi \to 180^{\circ }$ can be so severe that even weak reflections from the anechoic chamber’s absorber blocks are more powerful [19].

It is clear that for indoor on-body UWB ranging in general, the range errors remain generally unaffected for low ϕ values, while high outliers occur for $\phi \to 180^{\circ }$, and somewhere in between, a ϕ interval exists where a bias is introduced. However, the ϕ values for which the range error statistics change, and the extent of these changes depends on several factors. One factor is the distance of the tag antenna to the body. If the antenna is further away from the body, the direct path begins to be obstructed at higher ϕ values; thus the influence of creeping waves and, consequently, reflections occur at higher ϕ values [22].

A second factor is tag placement, wherein the influence on range errors and packet loss has been investigated in [21,31], respectively. Both works identified the head as the best place for the tag, experiencing almost no negative effects, and the chest/stomach as the worst place. To characterize the range error, [21] proposes a switch from a Gaussian function to another PDF, depending on the tag position, when a ϕ-like variable crosses a threshold.

Thirdly, the environment affects the range error statistics, as shown in our prior research [29], which consisted of static on-body UWB measurements in a lab and office environment. Although a similar orientation-dependent error pattern is observed in both environments, the distribution becomes skewed above $100^{\circ }$ in the office, while this only occurs above $140^{\circ }$ in the lab environment. This is attributed to a combination of increased tag–anchor distances lowering the signal-to-noise ratio and an increased amount of reflected paths, causing the UWB tag to often detect the reflected component instead of the diffracted component. On the other hand, the lack of reflective surfaces in the outdoor experiment in [22] allows for creeping waves to be dominant for all $\phi >90^{\circ }$. Fourthly, range estimation algorithms perform differently under HBS conditions as reported in [32], which compared the leading-edge detection algorithm with the SAGE algorithm. Lastly, given the fact that the creeping waves are heavily attenuated for high ϕ values, it is suspected that the transmit power also affects the range error distribution.

### 2.2. Detection and Mitigation of Human Body Shadowing Effects

UWB positioning in general NLoS conditions is usually performed in two steps. The first step is to identify the NLoS links in order to know which ranging measurements are likely to have large errors. This information is then used in the second step to mitigate position errors, of which the most common methods are weighing [26] or omitting [33] the range or estimating the range error and subtracting it from the measured range [34] before estimating the position.

Range-based NLoS detection and mitigation methods, e.g., [35], use subsets of available range measurements to estimate candidate positions, after which hypothesis testing on the range residuals of these positions is used to find NLoS tag–anchor links. These methods have the advantage of not requiring knowledge of the range error noise or CIR statistics; however, they require range estimates from several anchors simultaneously to estimate a position. This makes them less accurate for dynamic UWB-TWR localization, which uses a sequential range estimation scheme.

CIR-based NLoS detection relies on the fact that under LoS conditions, most of the signal’s energy is found in the first path component [36,37]. Some works derive features (e.g., kurtosis, excess delay spread, etc.) from the CIR and use statistical methods and machine learning algorithms, e.g., Pearson correlation [38], generalized Gaussian distribution [39], or support vector machines [40], to classify a measurement as LoS or NLoS. Other works employ deep learning techniques directly on the raw CIR data, of which the CNN is most popular method [34,36]. These CNNs are used as either a classifier to detect NLoS conditions [36] or as regressors to estimate the range error directly [34]. The biggest advantage of these machine learning and deep learning techniques is that they are parameterless, but they do require a substantial amount of training data. Also, a large part of experimental UWB-related research has been conducted with low-cost COTS UWB hardware, primarily using the Decawave DW1000 transceiver [32]. A disadvantage of the DW1000 is that reading the CIR from the device’s serial port is time consuming, which makes NLoS mitigation challenging in a dynamic setting. The authors of [37] proposed thresholding the difference between the total estimated power of the received signal and its first path component as a feature for NLoS detection. This feature is calculated from the DW1000’s metadata, which can be read much faster than the CIR. On top of that, NLoS conditions can be reliably detected by a single threshold [31,37]. However, it is shown for on-body localization scenarios that the orientation is a better feature for detecting HBS conditions, especially in a realistic environment [29].

While many works on general UWB NLoS detection and mitigation exist, few differentiate between different types of NLoS links and/or focus on HBS. The authors of [41] observed widely varying range error distributions depending on the obstructions, while [23] proposed orientation-aware range error distributions for on-body pedestrian tracking. In addition to the orientation-dependent error caused by HBS, the detection of the obstruction type can be used to mitigate NLoS more accurately. A fuzzy classifier was developed in [38], which labels a measurement as a combination of several NLoS types, including HBS. While a 50% reduction in the root-mean-squared localization error was achieved, [38] regarded hbs as merely humans obstructing the direct path and did not incorporate the relative orientation for on-body localization. Also, the localization algorithm in [38] needs range measurements from multiple anchors to estimate a position, which is not desirable for dynamic localization as previously mentioned. Solutions for dynamic (pedestrian) localization incorporate NLoS detection/mitigation strategies into tracking algorithms (i.e., filters), which estimate a new position for each new range measurement while taking previous measurements into account. State-of-the-art solutions have added IMUs to their pedestrian tracking algorithms [25,26,28]. By integrating the inertial data [25] or by employing a PDR algorithm [26,28], these systems combine inertial localization systems unaffected by NLoS conditions but prone to drift errors with absolute UWB positioning. On top of that, active NLoS detection was added based on the estimated walking direction and distance derived from IMU data in [25,26], but the orientation itself was not used for HBS mitigation. The authors of [28] do differentiate between spatial obstructions and HBS by respectively mapping the obstructions and by using an IMU to estimate the relative user orientation. By incorporating different mitigation strategies for each NLoS type into an adaptive Extended Kalman Filter (EKF), [28] improved the latter approach by 40%. Although good HBS mitigation results are achieved, [28] assumes a handheld tag that is 25 cm in front of the body. Consequently, the body-induced range error is Gaussian distributed with a constant standard deviation for all orientations, which does not grasp the full complexity of the problem. Also, in our opinion, an on-body tag allows for a wider range of real-world applications in, e.g., sports and industry, in which the athlete or worker already uses both hands.

Despite the abundance of research on UWB NLoS/HBS detection and mitigation, there is limited research on exploiting the knowledge of body orientation to mitigate HBS effects for on-body UWB localization. The relative orientation was used in [27], in which the orientation was estimated using a sequence of previously estimated positions. This was incorporated into a PF with an orientation-aware range error model as proposed in [23] to mitigate HBS effects. This orientation estimation strategy is flawed, as the orientation cannot be correctly estimated when the pedestrian is not walking in a straight line. In fact, ref. [27] reported an 82% reduction in the localization error with a chest-mounted tag when the ground truth orientation was used but only a 36% reduction when the estimated orientation was used. Furthermore, the orientation-aware range error model used in [27] considers all range measurements as LoS for orientations below a threshold, using a Gaussian range error model for its PF algorithm. Above the threshold, the channel is considered NLoS and the Gaussian (LoS) distribution is replaced by a Gamma distribution, where it is fitted on training data for which the true orientation is also above the threshold. However, the transition from a Gaussian-like (LoS) PDF to a heavy-tailed (NLoS) PDF can occur gradually and begin at varying angles [29], as this transition is affected by several factors, as discussed in Section 2.1. For these reasons, it is cumbersome to identify an optimal threshold for which the channel condition becomes NLoS, and the model does not fit well for range measurements with an orientation that is close to the threshold. Also, when the true orientation is close to the threshold, even small errors in the orientation estimation can cause the selection of the wrong distribution.

Therefore, we propose the use of mixture models as a better choice (as, e.g., in [17]), as they model distributions that are the sum of several unobserved variables. More specifically, we propose GMMs as they allow for the use of Kalman Filter (KF)-based algorithms, which are faster than their PF counterpart. Furthermore, we also use an IMU to estimate the tag heading more accurately as proposed in our previous work [29], and we make the tag–body orientation independent from the walking direction, unlike in [27]. The IMU can also be used to fuse our proposed algorithm with existing PDR methods as in [28], but this is beyond the scope of this work.

## 3. Materials and Methods

### 3.1. Experiment Setup

The experiment setup is divided into five parts. The first part explains the data collection. The next three parts cover the hardware itself, which consists of the UWB system, the IMU, and the ground truth system. The fifth part describes the measurement environment and trajectories. A summary of the hardware settings and experiment details is provided in Table 1.

#### 3.1.1. Data Collection

A Raspberry Pi (RPi) read both the UWB tag and IMU output using its serial ports. The mocap output as well as the data read by the RPi were published to a local MQTT broker. All sensor data were received in order and timestamped by subscribing to their MQTT topics.

#### 3.1.2. UWB

The UWB measurements were performed using “Wi-Pos” [42]. This hardware platform is based on the Decawave DW1000 UWB transceiver, which was controlled by a Zolertia RE-Mote. The latter orchestrated the ranging scheme using its CC1200 sub-GHz radio and was connected to the transceiver via a custom PCB. Each device could be configured either as tag or anchor, and all devices were equipped with an elliptical UWB patch antenna in this work. Symmetrical double-sided UWB-TWR was performed between the tag and one anchor at a time according to a time division multiple access protocol, after which the range was read from the tag’s serial port over USB.

#### 3.1.3. IMU

The IMU used in this work was the 9-Degrees of Freedom (DoF) Adafruit BNO055. It is composed of a 3-axis accelerometer, gyroscope, and magnetometer, which measure acceleration, angular rate, and magnetic field flux density, respectively, along three orthogonal axes. The BNO055 device also comes with a built-in AHRS, which fuses the data of these sensors in order to obtain absolute orientation. At the start of the experiment, the device is held still along each of its axes and must be rotated along two axes for calibration of the gyroscope and magnetometer. The sensor data (and calibration flags) were read over the BNO055’s UART interface at 100 Hz.

#### 3.1.4. Motion Capture (Ground Truth)

A Qualisys mocap system delivered mm-level accurate ground truth at 90 Hz, and it used its infrared (IR) cameras to track the IR markers that were attached to the carried setup. However, the human body makes it more difficult to consistently track markers on the UWB tag, especially near the edges of the capture area. Therefore, all hardware was taped onto a piece of cardboard and three markers were put on the sides, as shown in Figure 2a. This solution allows the mocap system to achieve a steady tracking rate of 90%. A rigid body was defined from the constellation of the IR markers using the mocap system’s user interface. Rigid bodies in the Qualisys mocap system have a local reference frame within, of which its origin is placed at the UWB antenna. The Z-axis points up, the Y-axis is perpendicular to the antenna surface and points away from the user (i.e., $\vec{PT}$), and the X-axis points to the user’s right, as shown in Figure 2a. In the remainder of the paper, the ’L’ and ’G’ subscripts denote local and global coordinate axes, respectively. Furthermore, in addition to 3D positions, the mocap system provides absolute orientation of the body in the form of a rotation matrix.

#### 3.1.5. Environment and Trajectories

Dynamic measurements were performed in the Industrial Internet of Things (IIoT) lab at IDLab, Ghent University. This lab has an 11 m × 9 m experiment area, equipped with eight Wi-Pos UWB anchors and seven Qualisys IR mocap cameras. One side of this environment is shown in Figure 2b, which is provided with close-ups of a UWB anchor and a mocap camera. Only four anchors, which were placed in a rectangular geometry at a height of 0.4 m, were used in our experiments for a more realistic anchor density given the relatively small area. The UWB anchors were attached to the walls with a distance of 15 cm between the walls and antennas. Two trajectories were designed, each performed five times, with each repetition taking on average 50 s to complete. The tag was held in front of the body at the abdomen while walking casually and while the arms were kept to the side, as shown in Figure 2a. Slight variations to the trajectories were conducted during each repetition.

The ground truth of one repetition of each trajectory is shown by the blue scatterplots in Figure 3a,b. The trajectories are outlined by red dashed lines, which mark the area in which mocap was able to track the body consistently. This area lies within the anchor geometry, shown by the blue dots in the corners of Figure 3a,b, and does not lie near the anchors. The geometric dilution of precision is therefore assumed to be fairly constant in the measurement area; thus, the UWB position error is only affected by the range error and not by the actual position. Also, the range error is not significantly affected by the true tag–anchor distance for the possible distances in this scenario [42]. Furthermore, no obstacles other than the person carrying the tag are present inside the anchor geometry. Therefore, the range error is only dependent on HBS effects.

Lastly, because our proposed work was based on range error PDFs, a training dataset was additionally created by randomly walking through the measurement area with the same hardware setup. This training dataset contains 4228 range measurements, for which all orientations are represented in almost equal quantity and with true tag–anchor distances in the (3,10) m range.

### 3.2. System Overview

Our proposed HBS mitigation approach is integrated into two filter algorithms, the KF and PF. This section provides a comprehensive, high-level system overview, which applies to both filters, as they are two realizations of the same concept.

#### 3.2.1. Range-Based Filtering

The filter algorithm tracks the state PDF of the tag worn by the pedestrian. The tracked state includes at least the tag’s position, as this is the desired output of the algorithm. Each time a new range measurement is available, the filter executes a prediction-update cycle. This is indicated by the loop in the green blocks in Figure 4, which shows a flowchart of the system. A new state is first predicted by propagating the state from the moment of the last measurement up to the moment of the new measurement. State prediction involves a process model $f_{x}$, which is a model of the dynamics of the tracked object and is also represented by a PDF. Assuming that the pedestrian walks casually along a flat trajectory, a simple Newtonian 2D Constant Velocity (CV) model is appropriate here. For this model, both the tag position and velocity are included in the estimated state and hence in the 4D vector (Equation 1) representing the state mean *x*, where x is the state PDF.
(1)$$x=E\left[x\right]={\left[\begin{array}{cccc} T_{x} & T_{y} & v_{x} & v_{y} \end{array}\right]}^{T}$$
*T* and *v* in (Equation 1) are the 2D tag position and velocity of the state mean, respectively.

The predicted state PDF ${\bar{x}}_{t+\Delta t}$ at time t+Δt is acquired by the convolution of the previous state PDF $x_{k}$ with the process model $f_{x}$, as described in (Equation 2) [43].
(2)$${\bar{x}}_{t+\Delta t}=x_{t}*f_{x}$$

The overline in (Equation 2) makes clear that this is the prior state PDF at time t+Δt. During the update step, the posterior state PDF $x_{t+\Delta t}$ is estimated by applying the Bayes theorem to incorporate the measurement. In the most general form, this involves multiplying the prior (i.e., $x_{t}$) by the likelihood L (i.e., the range error PDF) and normalizing the result [43], as in (Equation 3).
(3)$$x_{t+\Delta t}=||L\cdot {\bar{x}}_{t+\Delta t}\left|\right|$$

The difference between each filter type lies in how the state PDF, (Equation 2), and (Equation 3) are implemented, which is discussed in Section 3.4. Despite the implementation difference, one common operation of range-based tracking algorithms, usually regarded as the first step of the measurement update, is calculating one or more range residuals. A range residual *y* is the difference between the measured range *z* and the distance between a candidate position and relevant anchor, as described in (Equation 4).
(4)$$\begin{array}{cc} y & =z-h(x)\\ \; & =z-\sqrt{{\left(A_{x}-T_{x}\right)}^{2}+{\left(A_{y}-T_{y}\right)}^{2}+{\left(A_{z}-Z\right)}^{2}} \end{array}$$
h(x) is typically denoted as a non-linear function transforming the state mean to the measurement space [43,44,45]. $\left(A_{x},A_{y},A_{z}\right)$ is the 3D position of the anchor, whereas $\left(T_{x},T_{y}\right)$ is the 2D position of a candidate position for the tag. *Z* denotes the tag height, and its dissimilar symbol is deliberately chosen to emphasize that this is not part of the tracked state. Instead, *Z* is the average height measured by mocap over the entire trajectory.

Before the first cycle starts, the filter is initialized (Figure 4). In our work, a common part of each filter was initializing the state mean. Assuming no prior knowledge of the position, the Linearized Least Squares (LLS) multilateration algorithm [46] as implemented for 2D positioning in [29] estimates the initial tag position (mean) after receiving the first four range measurements. The mean velocity was set to $0\;\frac{m}{s}$, as it is assumed that the user is stationary at the start of the experiment. Lastly, to keep equations concise, the *t* and t+Δt subscripts were dropped. In the remainder of the paper, *x* and *P*, $\bar{x}$ and $\bar{P}$, and $x^{\prime }$ and $P^{\prime }$ denote the state mean and covariance at the start of the cycle, after the prediction step, and after the update step, respectively.

#### 3.2.2. Proposed HBS Detection and Mitigation Technique

Filter algorithms can accurately estimate the state while mitigating measurement errors by fusing information on past measurements and the dynamics of the tracked object. However, an important requirement is that the likelihood function matches the PDF of the actual (range) measurement errors. As discussed in Section 2.1, the range error PDF under HBS conditions is dependent on ϕ. This is where the yellow and red colored blocks in Figure 4 come into play.

We propose complementing the system with an IMU, of which the data are fused with the filter algorithm in order to estimate ϕ. As depicted in the yellow blocks of the flowchart, the estimated ϕ angle $\hat{\phi }$ is used directly to select the GMM-based range error model that is fitted for the current orientation. $\hat{\phi }$ itself is estimated by combining the filter’s predicted position with the (bias-corrected) IMU data along with the known anchor positions. Training the models requires an offline phase, in which mocap data are used to calculate the UWB range errors and their corresponding ground truth ϕ value ($\phi_{gt}$). This offline phase is depicted by the red blocks in Figure 4, encircled by a dashed line.

### 3.3. Characterization of Human Body Shadowing Effect on UWB Range Errors

This section describes the estimation of the ϕ angle and the proposed error model in more detail.

#### 3.3.1. Estimation of Tag–Body–Anchor Orientation $\hat{\phi }$

HBS effects are related to the body–tag orientation relative to the anchor. Given the vector definitions in Figure 1, $\phi \in [0^{\circ },180^{\circ }]$ is calculated by rewriting the scalar product of $\vec{PT}$ and $\vec{TA}$ for ϕ (Equation 5).
(5)$$\phi =\arccos\left(\frac{\vec{PT}\cdot \vec{TA}}{\left||\vec{PT}||\;||\vec{TA}|\right|}\right)$$
$\vec{TA}$ is estimated by a candidate 2D position of the tag and the known position of the anchor being ranged with. These positions are defined in the global coordinate frame $\left(X_{G},Y_{G}\right)$, which is aligned with the coordinate frame of the mocap system. To estimate $\vec{PT}$, we rely on the assumption from Section 3.1 that the local Y-axis $Y_{L}$ always points away from the body and is therefore equivalent to $\vec{PT}$. In that case, we can introduce the tag yaw $\theta_{tag}$ as the angular deviation of $\vec{PT}$/$Y_{L}$ from $X_{G}$, as illustrated in Figure 1. Given $\theta_{tag}$, calculating $\vec{PT}$ is straightforward (Equation 6).
(6)$$\vec{PT}=[\cos\left(\theta_{tag}\right),\sin\left(\theta_{tag}\right)]$$

To estimate $\theta_{tag}$, we employ the yaw $\theta_{imu}$ provided by the IMU’s built-in AHRS. However, $\theta_{imu}$ represents the angular deviation of the local X-axis $X_{L}$ from the north vector. Therefore, the global bias $b_{G}$, the deviation of the global X-axis $X_{G}$ from the north vector, has to be subtracted from $\theta_{imu}$. Furthermore, the local bias $b_{L}$, the deviation of the local X-axis $X_{L}$ from vector $\vec{PT}$, has to be subtracted from $\theta_{imu}$ too due to IMU placement. Thus, the relationship between $\theta_{tag}$ and $\theta_{imu}$ is described by (Equation 7).
(7)$$\theta_{tag}=\theta_{imu}-b_{G}-b_{L}$$

In this work, $Y_{G}$ aligns with north and $X_{G}$ with east (i.e., east, north, up reference frame), as illustrated in Figure 1, thus $b_{G}=-90^{\circ }$. As shown in Figure 1 and Figure 2a, $X_{L}$ points to the user’s right, thus $b_{L}=-90^{\circ }$ for this setup. In practice, these biases can be measured simultaneously by pointing $\vec{PT}$ to $X_{G}$. Because $\theta_{tag}$ should be $0^{\circ }$, the AHRS output is the total bias to be subtracted. This simple calibration comes in addition to the standard IMU calibration process, which is semi-automated in the Adafruit BNO055 system and of which the theoretical background can be found in the literature [47].

Note that this method is independent of the tag position on the body. In fact, if $b_{L}$ is constant, i.e., if the top of the IMU in Figure 2a is pointed away from the body, placing the equipment on another part of the body has no effect on how ϕ is estimated.

Lastly, the mocap system provides the tag position and a rotation matrix $R\in R^{3X3}$. The columns of *R* represent the axes of the local coordinate frame; thus, when taking Figure 1 and Figure 2a into account, the ground truth vector ${\vec{PT}}_{gt}=R_{1:2,2}$. Similarly, ${\vec{TA}}_{gt}$ is based on both the ground truth tag and anchor positions. The ground truth $\phi_{gt}$ is then calculated with (Equation 5) but using ${\vec{PT}}_{gt}$ and ${\vec{TA}}_{gt}$ instead. In the remainder of the paper, $\hat{\phi }$ and $\phi_{gt}$ represent the estimated and ground truth values of ϕ.

#### 3.3.2. Robust Orientation-Aware Gaussian Mixture Error Model

When incorporating a measurement, i.e., the measurement update in Figure 4, the filter’s likelihood function should resemble the PDF of the measurement error. In order to model this phenomenon more accurately than state-of-the-art models, we propose a robust range error model based on a bank of GMMs. For each integer value i∈[0,180], a GMM is trained on a subset of range errors from the training dataset, which is sampled around $\phi_{gt}=i$. Whenever a $\hat{\phi }$ value is estimated during the online phase, the GMM corresponding to $\phi_{gt}=\hat{\phi }$ is selected for the measurement update. This means a Gaussian-like distribution is selected for small $\hat{\phi }$ values (LoS), which transitions to a heavy-tailed PDF being selected when $\hat{\phi }\to 180^{\circ }$, i.e., in NLoS conditions.

When fitting the GMMs, our intention is to obtain a PDF that describes the range error PDF well for any ϕ but is general enough to work with $\hat{\phi }$ errors of several degrees. To sample a subset of range errors for a given ϕ, the errors are weighted based on their corresponding $\phi_{gt}$ value by a Gaussian window centered on ϕ. The weight $w_{\phi_{gt}}$ of a range error with corresponding $\phi_{gt}$ in the dataset $S_{err}$ for a given ϕ is calculated with (Equation 8), in which $\phi_{gt}^{\left(k\right)}$ is the $\phi_{gt}$ value corresponding to range error *k*.
(8)$$w_{\phi_{gt}}\left(\phi \right)=\frac{\exp\left(\frac{{\left(\phi -\phi_{gt}\right)}^{2}}{-2\sigma^{2}}\right)}{\sum_{k\in S_{err}}\exp\left(\frac{{\left(\phi -\phi_{gt}^{\left(k\right)}\right)}^{2}}{-2\sigma^{2}}\right)}$$

A random subset of range errors is then selected by multinomial resampling, in which range errors with a higher weight have a higher chance of being selected. Thus, most selected errors correspond with a $\phi_{gt}$ that is close to ϕ, with some corresponding to a $\phi_{gt}$ that is further from ϕ depending on the selected σ.

The Expectation-Maximization (EM) algorithm as implemented in [48] is used in this work to fit the GMMs. As the GMM is a weighted sum of Gaussians, the EM algorithm’s main hyperparameter is the amount of Gaussian components to fit. To find the ideal amount, a sequence of GMMs are fitted for each subset, of which the first GMM has one Gaussian component, the second has two components, etc. The GMM with the lowest Bayesian Information Criterion (BIC) value is then selected for each subset, resulting in a bank of 181 GMMs, each having $K_{\phi }$ components. Thus, given the orientation ϕ and range residual *y* for a candidate position, the measurement likelihood function L is described by (Equation 9).
(9)$$L\left(\phi ,\;y\right)=\sum_{k=1}^{K_{\phi }}\pi_{\phi }^{\left(k\right)}\cdot N\left(y:\mu_{\phi }^{\left(k\right)},\;\sigma_{\phi }^{2(k)}\right)$$

The $3K_{\phi }$ parameters in (Equation 9) are calculated by the EM algorithm [48], where $\pi_{\phi }^{\left(k\right)}$ is the weight of the *k*-th out of $K_{\phi }$ Gaussian components of the GMM with index ϕ with the lowest BIC value. This offline phase is denoted by the red blocks in Figure 4.

### 3.4. Mitigation of Human Body Shadowing Effects

HBS effects on UWB localization are being mitigated by combining the IMU-based HBS characterization approach from Section 3.3 with a tracking algorithm, which causes the latter to adapt its measurement noise model to the input data. So far, the roles of the relative orientation ϕ and GMM-based error model in the localization system have been discussed in Section 3.2, and how they are acquired has been discussed in Section 3.3. However, the evaluation of HBS effects relies on the IMU as well as the estimated position (Equation 5). The effectiveness of the proposed HBS mitigation strategy is therefore highly dependent on the positioning algorithm itself. In this section, the implementation of our work for two known filter variants, the PF and Unscented Gaussian Sum Filter (UGSF) algorithms, is discussed in more detail.

Furthermore, filters are designed for real-time tracking as they incorporate past measurements up to the latest one to estimate a new position. However, for applications where a certain delay is permitted, smoothers can improve accuracy by also including future measurements. Therefore, the effect of our proposed HBS mitigation technique on smoothers is also explored.

#### 3.4.1. Unscented Gaussian Sum Filter

The GSF or Gaussian Mixture Filter (GMF) is similar to the KF and its variants. While the state, process model, and measurement error model are represented by Gaussian PDFs in the KF, they are represented by GMMs in the GSF [44]. For each Gaussian component, the GSF employs the EKF [44] or Unscented Kalman Filter (UKF) [49] equations. This enables the GSF to handle non-Gaussian processes and/or measurement noise while still being able to exploit the efficiency of the KF’s closed-form solution.

Our filter is initialized with a one-component GMM, i.e., a single Gaussian PDF, which is fully described by its mean vector (Equation 10) and covariance matrix (Equation 11).
(10)$$\begin{array}{cc} \; & x={\left[\begin{array}{cccc} T_{x,0}^{LLS} & T_{y,0}^{LLS} & 0 & 0 \end{array}\right]}^{T} \end{array}$$
(11)$$\begin{array}{cc} \; & P=\left[\begin{array}{cccc} \sigma_{T,0}^{2} & 0 & 0 & 0\\ 0 & \sigma_{T,0}^{2} & 0 & 0\\ 0 & 0 & \sigma_{v,0}^{2} & 0\\ 0 & 0 & 0 & \sigma_{v,0}^{2} \end{array}\right] \end{array}$$

As discussed in Section 3.2, the initial position is estimated using the LLS algorithm, and the user is assumed to be stationary. $\sigma_{T,0}^{2}$ and $\sigma_{v,0}^{2}$ are the initial noise variances of the position and velocity, which are each set as equal for both axes. The process model is linear with Gaussian noise; thus, the prediction step is performed using the standard KF Equation (Equation 12) in this work.
(12)$$\bar{x}=N\left(\bar{x},\;\bar{P}\right)=N\left(Fx,\;{\mathrm{FPF}}^{T}+Q\right)$$

The state transition matrix *F* of the CV process model and process noise Q are provided in (Equation 13) and (Equation 14), respectively.
(13)$$\begin{array}{cc} \; & F=\left[\begin{array}{cccc} 1 & 0 & \Delta t & 0\\ 0 & 1 & 0 & \Delta t\\ 0 & 0 & 1 & 0\\ 0 & 0 & 0 & 1 \end{array}\right] \end{array}$$
(14)$$\begin{array}{cc} \; & Q=\left[\begin{array}{cccc} \frac{\Delta t^{4}}{4} & 0 & \frac{\Delta t^{3}}{2} & 0\\ 0 & \frac{\Delta t^{4}}{4} & 0 & \frac{\Delta t^{3}}{2}\\ \frac{\Delta t^{3}}{2} & 0 & \Delta t^{2} & 0\\ 0 & \frac{\Delta t^{3}}{2} & 0 & \Delta t^{2} \end{array}\right]\cdot \sigma_{w}^{2} \end{array}$$
Q is derived from the piecewise white noise model with acceleration being the highest order term [43], where $\sigma_{w}^{2}$ is the process noise variance. Thus, after prediction, the mean of the Gaussian state PDF has shifted in the direction of the velocity vector, and the covariance has increased due to the process noise. Given the predicted state mean $\bar{x}$ and the concurrent calibrated AHRS measurement, $\hat{\phi }$ is calculated using (Equation 5) and (Equation 6), in which $\bar{x}$ is substituted as tag position *T*. As discussed in Section 3.3, the measurement error model for a given $\hat{\phi }$ is a GMM with $K_{\hat{\phi }}$ components (Equation 9). Therefore, after the measurement update, the state PDF $x^{\prime }$ is a GMM with $K_{\hat{\phi }}$ components (Equation 15), where ${w^{\prime }}_{\hat{\phi }}^{\left(k\right)}$, ${x^{\prime }}^{\left(k\right)}$, and ${P^{\prime }}^{\left(k\right)}$ are the weight, mean, and covariance of *k*-th component of the updated state, respectively.
(15)$$x^{\prime }=\sum_{k=1}^{K_{\hat{\phi }}}{w^{\prime }}_{\hat{\phi }}^{\left(k\right)}\cdot N\left({x^{\prime }}^{\left(k\right)},{P^{\prime }}^{\left(k\right)}\right)$$

Each Gaussian component $N({x^{\prime }}^{\left(k\right)},{P^{\prime }}^{\left(k\right)})$ of the posterior (Equation 15) is estimated by a separate Kalman-type filter [44]. Each *k*-th filter in this work is initialized with the same prior (Equation 12) and performs the measurement update using the *k*-th Gaussian component of the selected GMM as the likelihood function. In this work, the UKF is chosen for the filter bank, hence the name UGSF, as first proposed in [49]. The UKF itself is a variant of the KF that is designed to handle non-linearities, e.g., the transform function h() in (Equation 4), by applying the Unscented Transform (UT) instead of an erroneous linearization step as in the EKF. Technical details on the UKF can be found in [50]. The weight ${w^{\prime }}_{\hat{\phi }}^{\left(k\right)}$ of each component is calculated with (Equation 16) [44], where *y* is the pre-fit residual and $P_{z}$ represents the state covariance in the measurement space. *y* is calculated by substituting $\bar{x}$ in (Equation 4). $P_{z}$ is a scalar, and is calculated by applying h() to the sigma points of one of the UKFs and calculating the weighted covariance, i.e., by applying the UT [50].
(16)$${w^{\prime }}_{\hat{\phi }}^{\left(k\right)}=\frac{\pi_{\hat{\phi }}^{\left(k\right)}\cdot N\left(y:\mu_{\hat{\phi }}^{\left(k\right)},P_{z}+\sigma_{\hat{\phi }}^{\left(k\right)}\right)}{\sum_{j=1}^{K_{\hat{\phi }}}\pi_{\hat{\phi }}^{\left(j\right)}\cdot N\left(y:\mu_{\hat{\phi }}^{\left(j\right)},P_{z}+\sigma_{\hat{\phi }}^{\left(j\right)}\right)}$$

Lastly, the state could be used as is for the next cycle. Each component would then be propagated as in (Equation 12) and then updated by a bank of UKFs as in (Equation 15). The latter would make the amount of Gaussian components in the state rise exponentially. Several approaches are mentioned in [44,49] to solve this problem. However, we chose to collapse the state back into a single Gaussian at the end of each cycle, using (Equation 17) and (Equation 18) [45].
(17)$$\begin{array}{cc} \; & x^{\prime }=\sum_{k=1}^{K_{\hat{\phi }}}w_{\hat{\phi }}^{\left(k\right)}{x^{\prime }}^{\left(k\right)} \end{array}$$
(18)$$\begin{array}{cc} \; & P^{\prime }=\sum_{k=1}^{K_{\hat{\phi }}}w_{\hat{\phi }}^{\left(k\right)}\left({P^{\prime }}^{\left(k\right)}+\left({x^{\prime }}^{\left(k\right)}-x^{\prime }\right){\left({x^{\prime }}^{\left(k\right)}-x^{\prime }\right)}^{T}\right) \end{array}$$

Simultaneously, (Equation 17) is used as the position output after each cycle.

#### 3.4.2. Particle Filter

In the PF, the state is represented by a set of particles.

At initialization, a set of *N* particles is sampled from a prior PDF, which can be of any type. Without prior knowledge of the state, the uniform distribution would be appropriate. However, to stay in line with the the GSF, the initial particle set is sampled from the same PDFs, (Equation 10) and (Equation 11), as is used for the initial state of the UGSF. Furthermore, each particle *p*’s state includes a weight $w_{p}$.

During prediction, a random Gaussian value is added to each particle’s velocity (Equation 19), after which the particles are propagated (Equation 20).
(19)$$\begin{array}{cc} \; & {\bar{v}}_{p}=v_{p}+w\in N\left(0,\;\sigma_{v}^{2}\cdot I_{2}\right) \end{array}$$
(20)$$\begin{array}{cc} \; & {\bar{T}}_{p}=T_{p}+\Delta t\cdot {\bar{v}}_{p} \end{array}$$

Next, the range residual $y_{p}$ is calculated and the orientation ${\hat{\phi }}_{p,t+\Delta t}$ is estimated for each particle *p* at time t+Δt by substituting the particle position $T_{p}$ as the candidate position in (Equation 4) and (Equation 5). Each particle is then reweighted according to (Equation 21), in which L is the likelihood function (Equation 9), i.e., the measurement noise model.
(21)$${w^{\prime }}_{p}=w_{p}\cdot L\left({\hat{\phi }}_{p},\;y_{p}\right)$$

The estimated position *T* at time t+Δt is estimated as the weighted average of the particle positions. Sampling importance resampling is applied to solve the known particle degeneracy problem, in which a new set of *N* particles is sampled from the updated particle set. In this work, this resample step is applied each cycle using stratified resampling. Finally, the resampled particle set becomes the prior PDF for the next cycle.

#### 3.4.3. Computational Efficiency Improvements

The PF’s major drawback is its computational complexity in both time and space. For example, a PF is several orders of magnitude slower than an EKF or UKF, depending on the amount of particles used. In this work, two improvements have been made to reduce the computation time of the proposed PF algorithm.

First, computing the likelihood (Equation 9) for each particle is time consuming, while the input is often very similar. Therefore, a lookup table (LUT) is constructed for each of the 181 GMMs. For each LUT, the stored keys are range error values in the range [−3,7] m with a resolution of 1 cm. The LUT values are the corresponding likelihoods that are calculated with (Equation 9). In the PF online phase, range residuals are rounded to 1 cm before the likelihood is retrieved from the LUT.

Second, because the process model is linear with Gaussian noise, the closed-form KF solution (Equation 12) can be used. Particles are then sampled from the Gaussian state PDF after the prediction step. After weighting the particles with the (LUT-based) likelihood function, the state is transformed back into a Gaussian PDF with (Equation 22) and (Equation 23).
(22)$$\begin{array}{cc} \; & x^{\prime }=\sum_{p=1}^{N}{w^{\prime }}_{p}\cdot {x^{\prime }}_{p} \end{array}$$
(23)$$\begin{array}{cc} \; & P^{\prime }=\sum_{p=1}^{N}{w^{\prime }}_{p}\cdot \left({x^{\prime }}_{p}-x^{\prime }\right){\left({x^{\prime }}_{p}-x^{\prime }\right)}^{T} \end{array}$$

#### 3.4.4. Smoothing

Applications that do not rely on real-time tracking can use all measurements (i.e., batch smoothing) to estimate each position. Alternatively, using the measurements up to current time *T* can be used to estimate the position at time T−L (i.e., fixed-lag smoothing), where *L* is the lag. A Rauch-Tung-Striebel (RTS) smoother [51] is implemented on top of the UGSF, which is denoted as RTS-UGSF. The PF is additionally implemented as a Backtracking Particle Filter (BPF) [52], in which the particle state is expanded with a reference to the particle from which it descends.

Both smoothers are implemented as fixed-lag smoothers in order to investigate the effect of the introduced lag on the localization accuracy. Therefore, as depicted in Figure 4, the *L* most recent estimated states are stored in memory. The BPF searches through the *L* latest generations of surviving particles by recursively looking up each particle’s predecessor *L* times. This results in a subset of the original particles at time T−L, which has surviving descendants at time *T*. The weights of this particle subset are normalized, and the smoothed state is then estimated as the weighted average of the subset.

The implementation of the RTS smoother depends on the filter type. Because the state is collapsed to a single Gaussian after each cycle, the RTS equations for the standard KF can be used as described [43]. Similar to the BPF, the RTS smoother begins at the latest state and works its way back to the state at time T−L.

## 4. Results

To evaluate the range errors, the first ground truth positions before and after each range measurement were interpolated. Range errors were then calculated by substituting the interpolated 3D ground truth tag position as $\left(T_{X},T_{Y},Z\right)$ in the residual Equation (Equation 4).

For each estimated position (i.e., for each UWB range measurement), the localization error was defined as the 2D Euclidean distance between this position and the interpolated ground truth position. Furthermore, each PF configuration was run ten times due to the randomness of the resampling step, i.e., each PF algorithm was run 100 times in total. The localization results of each algorithm discussed in this section were calculated on the union of localization errors of all runs. Because the mocap tracking rate is 90%, the range and position errors were only calculated if the time between the two mocap positions to be interpolated was less than 0.1 s.

### 4.1. Human Body Shadowing Effect on UWB Ranging Accuracy

Figure 5 shows four percentile errors as a function of $\phi_{gt}$. These percentile errors were calculated using the same subsets as for fitting the GMMs in Section 3.3. All percentiles are quite constant for $\phi_{gt}<80^{\circ }$ and only slightly increase when $\phi_{gt}$ approaches $90^{\circ }$. The p99 error of 32 cm and median error of 4 cm are in line with results from off-body measurements with the same hardware in the same environment [42]. This confirms the findings of other works discussed in Section 2.1 that UWB ranging is not significantly affected by the human body when the tag antenna is in visible LoS of the anchor antenna. The range error clearly increases with $\phi_{gt}$ when $\phi_{gt}>90^{\circ }$. The median error increases to 32 cm for $\phi >160^{\circ }$, which equals the p99 error for $\phi_{gt}<90^{\circ }$. The p99 error for $\phi >160^{\circ }$ is 1.53 m, with outliers reaching up to 4.05 m. Thus, with the growing (amount of) outliers for increasing $\phi_{gt}$, the range error distribution becomes more skewed.

This is reflected in Figure 6, which shows the range error histograms in blue for four $\phi_{gt}$ values, with the fitted GMMs in red. Figure 6 clearly shows how the optimal amount of Gaussian components needed to fit a GMM to the range error subset increases monotonically with $\phi_{gt}$ due to the increasing skewness. This monotonic increase also occurs with the range error that corresponds with the maximum of the GMM. The GMM in Figure 6a ($\phi_{gt}=60^{\circ }$) has two Gaussian components. Having similar means but strongly divergent variances, these components form an almost symmetrical Gaussian-like PDF, which simultaneously has a narrow peak at 2 cm and elongated tails. Figure 6b ($\phi_{gt}=100^{\circ }$) shows that the distribution has shifted, with its maximum being at 4 cm. The left tail has shrunk, while a heavier right tail has appeared, indicating that one component now has a larger mean as well as a larger variance. At $\phi_{gt}=150^{\circ }$, the range bias caused by creeping waves has shifted the peak of the distribution to 12 cm. Range errors caused by reflections also start to occur, which causes the PDF to become heavily skewed. Due to an increased amount of range errors caused by reflections, a large part of the errors in Figure 6d ($\phi_{gt}=150^{\circ }$) lie outside of the main peak (30 cm), almost forming a second mode around 70 cm.

### 4.2. Human Body Shadowing Mitigation for UWB Localization

#### 4.2.1. Algorithm Configurations

Abbreviations are used in this section for a concise discussion of the various localization algorithms. Our proposed HBS mitigation approach was implemented on top of a PF and UGSF as described in Section 3, which is denoted here as PF-Prop and UGSF-Prop. We also implemented a state-of-the-art PF algorithm (PF-Ref) [27] referenced in Section 2. We used its Gaussian gamma range error model and heading estimation algorithm to analyze the impact of our proposed IMU-based orientation estimation (PF-Mixed-IMU) and GMM-based error model (PF-Mixed-GMM) separately. Furthermore, we also implemented a UKF with orientation-aware anchor selection (UKF-AS), in which the update step is skipped when $\hat{\phi }>\phi_{c}$, where $\phi_{c}$ is a fixed threshold. This is performed to verify whether using all measurements with the proposed orientation-aware range error model is better than simply omitting measurements under high HBS influence. For the dataset used in this work, the optimal threshold is $\phi_{c}=125^{\circ }$, which is used for all following results of UKF-AS. The evaluated filter algorithms and their smoother variants are summarized in Table 2.

#### 4.2.2. Benchmark of Proposed Algorithms

Our proposed algorithms are compared with some well-known localization algorithms (LLS, EKF, UKF), as well as a state-of-the art HBS mitigation algorithm (PF-Ref) from [27]. The EKF has a median and p99 error of 23 cm and 73 cm, respectively, and is used as the main benchmark algorithm. Thus, improvements in this section are expressed in percentages relative to the standard EKF unless specified otherwise. Figure 7 shows the Cumulative Distribution Function (CDF) of localization errors for the discussed algorithms. BPF-Prop performs the best, reducing the median and p75 error to 12 cm (−43%) and 18 cm (−45%), respectively. While achieving very high accuracy, this algorithm does not provide real-time results. In fact, the lag selected for the smoother algorithms is 0.6 s. Therefore, the best performing real-time algorithms are (k)PF-Prop and UGSF-Prop, which outperform all other algorithms. kPF-Prop and UGSF-Prop achieve a median error of 13 cm (−38%) and 15 cm (−29%) and a p99 error of 46 cm (−37%) and 48 cm (−34%), respectively. kPF-Prop is a faster variant of the PF-Prop and has the same performance as PF-Prop. PF-Ref is outperformed by our proposed algorithms, achieving a median and p99 error of 19 cm (−10%) and 58 cm (−21%), respectively. When not fitting the PDFs of PF-Ref to our training dataset but using the PDF parameters fitted for the experiments in [27] instead, the resulting PF-Ref (unfit) diverges. Furthermore, Figure 7 shows our proposed algorithms perform better than simply omitting the measurements under HBS influence, as performed in UKF-AS. More specifically, the p90 error of UGSF-Prop is 24% lower than that of UKF-AS.

UWB localization experiments in LoS conditions, in which the tag was placed on top of a moving cart, have been performed in the same environment with identical UWB and mocap infrastructure in [18], although eight UWB anchors were used. The EKF in the LoS experiments achieved a p50 and p90 error of 5 cm and 10 cm, respectively. We tested the EKF and our proposed KF on the same eight anchors for a fair comparison. The EKF achieves a p50 and p90 error of 16 cm (+320%) and 31 cm (+210%), respectively, while the PF-Prop achieves 9 cm (+180%) and 20 cm (+100%), respectively. Thus, our proposed PF algorithm does not completely remove the HBS effects on localization errors but does mitigate these effects significantly. Figure 8 shows red scatter plots of the two trajectories as estimated by the EKF (Figure 8a,e; Figure 8b,f; Figure 8c,g; and Figure 8d,h). The ground truth is represented by the blue scatter plots. The improved performance of the referenced PF compared with the EKF and proposed PF compared with the referenced PF and EKF is clearly visible. BPF-Prop, with a lag of 0.6 s, overlaps nicely with the ground truth trajectory.

#### 4.2.3. Selecting Algorithm Parameters

In previous sections, several parameters are mentioned that affect the performance of the discussed positioning algorithms. The values of some important parameters are motivated in this section.

An important parameter is the amount of particles that is used by (variations of) the PF algorithm. While more particles generally means better performance, large amounts of particles can increase the computational burden to a point where, depending on the hardware, real-time positioning is no longer possible. Figure 9a shows the average localization error of the PF variations as a function of the amount of particles (*N*) used. The average localization error of all PFs decreases sharply for each added particle for N<150 and keeps decreasing steadily until N=400. For N>400, many more particles are needed for limited performance gain.

Figure 9b shows the average localization error of the smoother variants of the proposed and referenced PF algorithms as a function of the fixed lag *L*. A lag of 0 s represents the real-time performance. The average error of each algorithm decreases sharply with increasing lag for L≤0.6 s.The highest performance is achieved by BPF-Prop, which has an average error of 11 cm at L=0.6 s. This is a 30% reduction in the average localization error compared with its real-time counterpart (PF-Prop). kPF-Prop cannot keep track of its particles’ predecessors because it collapses the state into a Gaussian after every measurement update. Therefore, it is combined with the RTS-smoother (RTS-kPF-Prop), achieving an average error of 12 cm (−20%). Thus, while PF-Prop is not as computationally efficient as kPF-Prop, its smoother variant performs significantly better.

#### 4.2.4. Contribution of the IMU- and GMM-Based Error Model

This section analyzes the contribution of the IMU-based HBS detection method and GMM-based error model separately. To do this, we implemented two PF variants (PF-Mixed-IMU and PF-Mixed-GMMr), which are a mix of our proposed PF and the reference PF [27]. PF-Mixed-IMU uses the range error model of PF-Ref but uses the heading provided by the IMU to switch between the Gaussian and Gamma PDFs. PF-Mixed-GMMr uses our proposed GMM-based model but uses the heading algorithm of PF-Ref. The results of these two variants are compared with PF-Ref and PF-Prop, of which the CDFs are shown together in Figure 10a. It is clear from Figure 10a that the PF-Mixed-GMMr does not improve the performance compared with the reference PF. This is explained by Figure 10b, which shows the discussed algorithms’ CDFs of absolute $\hat{\phi }$ errors. The referenced position-based heading algorithm [27,53] performs worse than the IMU-based heading algorithm, which causes a worse estimation of $\hat{\phi }$. Because of high $\hat{\phi }$ errors, a badly matching PDF is often selected. The IMU-based heading of the PF-Mixed-IMU, which uses the referenced range error model, improves the performance because the (more) correct PDF is selected more often. While the IMU-based $\hat{\phi }$ estimation is not free of errors ($p95=17^{\circ }$), its improved performance in combination with our proposed range error model (PF-Prop) delivers the highest accuracy.

#### 4.2.5. Runtime Analysis

This section investigates the potential of our proposed algorithms for use on battery-powered, resource-constrained Internet of Things (IoT) devices. For this reason, a runtime analysis is performed on an RPi 4 model B running Python 3.11. The presented runtimes are the averages of running each algorithm ten times on one recorded trajectory with a duration of 53 s or 624 UWB measurements. Figure 11 shows the runtimes of the proposed filter algorithms compared with the UKF. The provided runtimes are relative to the slowest algorithm, i.e., PF-Prop, and are shown on a logarithmic scale. As commonly known, the UKF is very efficient with a runtime of only 0.9 s the RPi 4, which is less than 1% of PF-Prop. The vertical dashed line across Figure 11 shows the duration of the recorded trajectory used for the runtime analysis. PF-Prop takes, on average, 98.1 s on the RPi 4 to compute a trajectory using 400 particles. This makes the initial version of our proposed PF unusable for real-time IoT applications. However, when using LUTs for calculating the likelihood (Equation 9) (PF-Prop-LUT), computation time is more than halved (47.2 s), making it suitable for real-time applications. PF-Prop, like the generic PF algorithm, iterates over all particles during the prediction and update step. On top of using LUTs, kPF-Prop-LUT uses the closed-form KF equations for the prediction step, only iterating over the particles during the update step. This further improves the proposed algorithm’s efficiency, resulting in a computation time (23.1 s) that is roughly half the duration of the original PF-Prop. Still, kPF-Prop’s computation time is roughly four times higher than that of UGSF-Prop (5.4 s) while only performing marginally better.

## 5. Discussion

### 5.1. Results

NLoS conditions still pose a difficult problem for accurate UWB positioning systems. In the UWB localization of pedestrians, human body shadowing is an important type of NLoS in which the body obstructs the direct path between an anchor and the on-body tag. It has been established in earlier works that the distribution of UWB range errors is highly dependent on the relative orientation of the body and tag with respect to the anchor. However, there is limited research on exploiting this knowledge to mitigate the effects of HBS-induced range errors on localization. After a thorough literature study (Section 2), this work proposed an IMU-based HBS mitigation approach for UWB localization. Our proposed method is implemented on top of two known tracking algorithms (Section 3), i.e., a PF and UGSF. The algorithms are evaluated on measurements recorded in the open area of an industrial lab environment with mm-level ground truth accuracy as provided by a mocap system (Section 4). The results are compared with those of a state-of-the-art algorithm and with several other commonly used tracking algorithms. Both the proposed PF and UGSF methods outperform all other algorithms, with the PF variant achieving the highest accuracy. However, the UGSF variant is more than ten times faster than the PF, which makes the former more suitable for constrained, low-power devices for, e.g., Internet of Things applications. Furthermore, we analyzed how our proposed method performs when combined with smoother algorithms. It is shown that for a delayed output of only two seconds, the PF implemented as a backtracking PF can further improve the accuracy significantly. Meanwhile, the UGSF implemented as an RTS smoother (with fixed delay), achieves a modest improvement compared with its real-time counterpart.

### 5.2. Limitations and Shortcomings

The experiments were performed in an open area with no obstacle other than the user being localized. Furthermore, the user walked at a relatively constant speed and did not make any sudden or diverse movements, e.g., quick turns, jumps, crouching, etc.

In a realistic environment, magnetic materials locally distort the Earth’s magnetic field, which affects the accuracy of the yaw angle that is estimated by the IMU [54]. As shown in Section 4.2.4, our proposed orientation-aware range error model performs better than the state of the art, but only when the yaw angle is accurately estimated. As such, it might be necessary to employ an algorithm that can detect these anomalies as proposed in [55].

In a realistic environment, other types of NLoS conditions may occur that are not taken into account in this work, e.g., concrete walls, metal racks, etc. This means that the range error distribution will not only depend on the orientation, but also on the position of the pedestrian. A map-based range error model could be combined with our current model, which takes into account static obstacles in the environment, similar to the two-step error correction algorithm proposed in [28].

In a scenario with multiple pedestrians, each pedestrian can obstruct the direct path between an anchor and any other pedestrian. Therefore, the tracking algorithm for each pedestrian would need access to the estimated states of the other pedestrians, which can then be used similarly to a map-based model.

Because the IMU yaw is independent of the walking direction, the correct distribution of the range error model can still be selected when the pedestrian makes sharp turns or walks sideways or backwards, although this has not been explicitly tested in this work. However, the localization performance might degrade when the pedestrian makes sudden turns or accelerations because of the constant velocity model that is used in the filter algorithms. Using a constant acceleration model would allow the localization system to respond faster to these events but would also make it more susceptible to outliers in the ranging measurements. Furthermore, more complex movements, such as jumping or crouching, have also been shown to affect the range error distribution as well as the rate of failed range measurements [31]. Lastly, the effect of different tag positions has been modeled before [23]. However, alternative tag positions can also be modeled by our proposed GMM-based model as the algorithm used for training our model does not make assumptions on the specific tag position.

### 5.3. Future Work

In light of the addressed limitations in Section 5.2, our future work will involve an evaluation of our proposed algorithms with more realistic movements and tag positions. Furthermore, we plan to extend our orientation-aware range error model to incorporate static obstructions as well as other pedestrians being localized. Lastly, similar to [56], we plan to expand our algorithm to the self-learning of the error model.

## Figures and Tables

**Figure 1 sensors-23-08289-f001:**
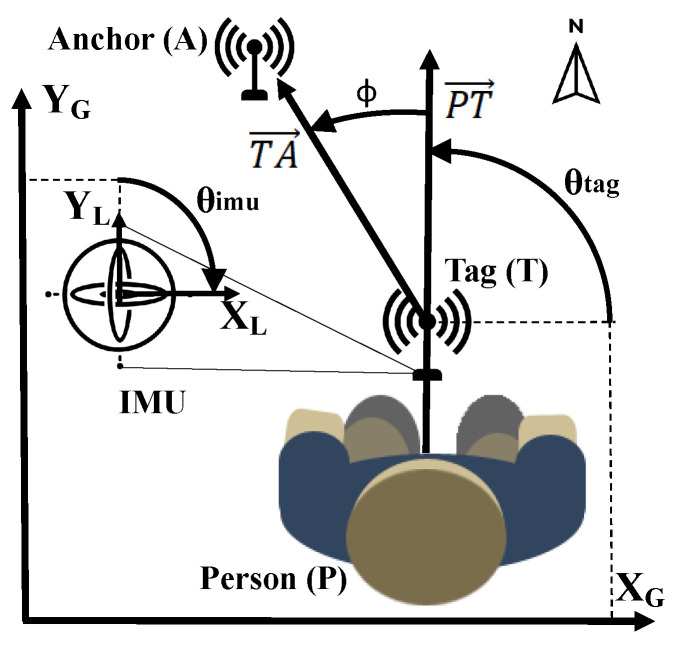
Visualization of angles and vectors related to estimation of the relative body orientation.

**Figure 2 sensors-23-08289-f002:**
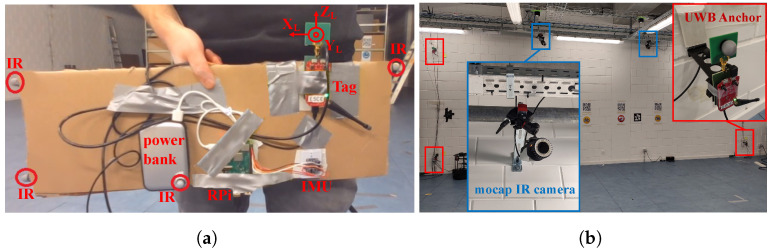
Part of the measurement area of the IIoT lab (**a**) and the on-body setup carried by the user (**b**).

**Figure 3 sensors-23-08289-f003:**
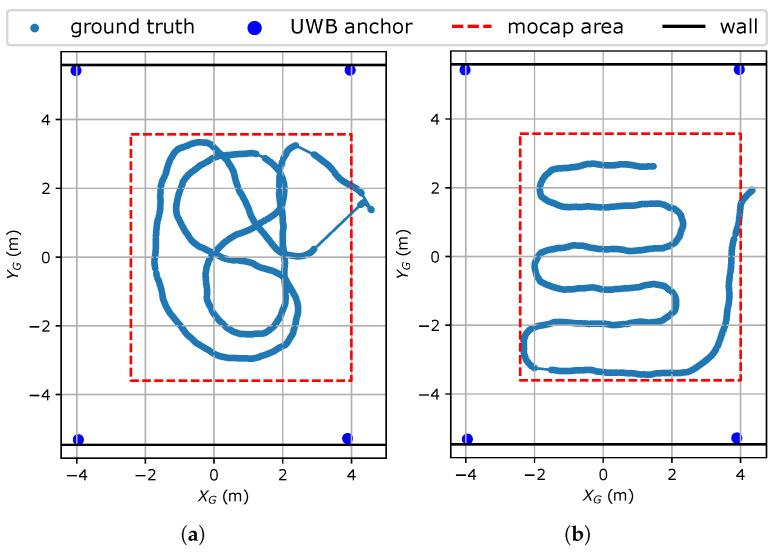
Ground truth trajectories, surrounded by a red dashed rectangle marking the effective measurement area. One trajectory (**a**) consists almost entirely of smooth turns, while the other (**b**) consists of straight parts and sharp turns. The dots in the corners represent the UWB anchors. Black lines represent concrete walls to which the anchors are attached.

**Figure 4 sensors-23-08289-f004:**
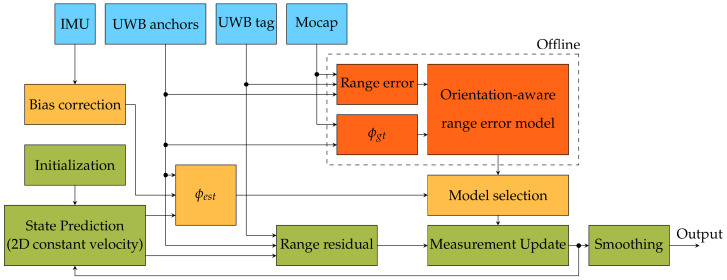
Flowchart of the human body shadowing mitigation approach. The blue blocks represent the hardware measurements and known UWB anchor locations. The green blocks represent the processes of a typical tracking algorithm. The red (encircled with dashes) and yellow blocks represent the offline and online part of the proposed mitigation method, respectively. The arrows show how the output of each block serves as input to one or more other blocks.

**Figure 5 sensors-23-08289-f005:**
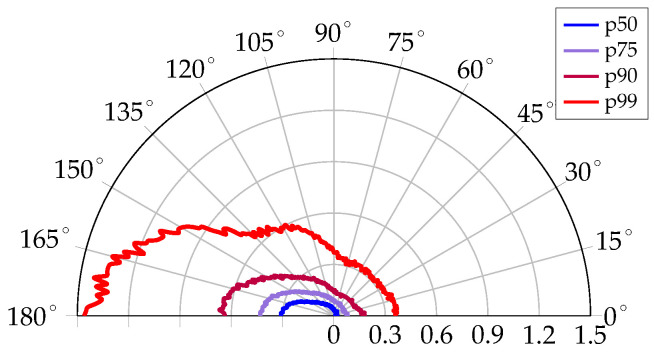
Range error statistics as a function of the body–tag–anchor orientation $\phi \in [0^{\circ },180^{\circ }]$.

**Figure 6 sensors-23-08289-f006:**
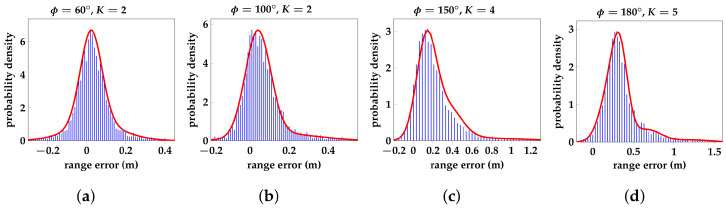
Histograms (blue) of UWB range error subsets with fitted GMMs (red). Each subset is sampled around ϕ with a Gaussian window from the IIoT lab static experiment dataset discussed in [29]. (**a**–**d**) show how the amount of Gaussian components *K* needed to train the model, is proportional to ϕ.

**Figure 7 sensors-23-08289-f007:**
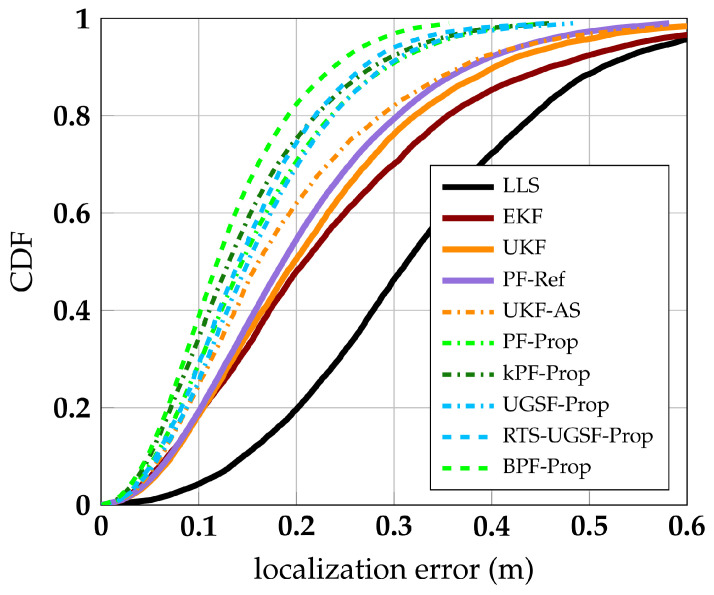
CDFs of localization errors of proposed and referenced algorithms.

**Figure 8 sensors-23-08289-f008:**
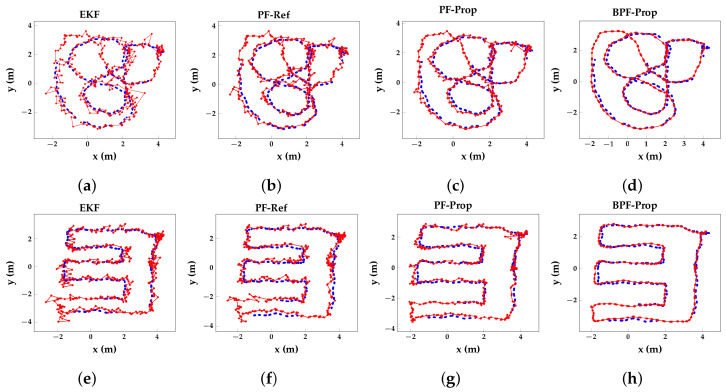
Plots of the estimated (red) over ground truth (blue) 2D trajectories for the EKF (**a**,**e**) algorithm, the reference [27] (**b**,**f**), the proposed filter algorithms (**c**,**g**), and the proposed smoother algorithm (**d**,**h**) with a delay of four UWB measurements.

**Figure 9 sensors-23-08289-f009:**
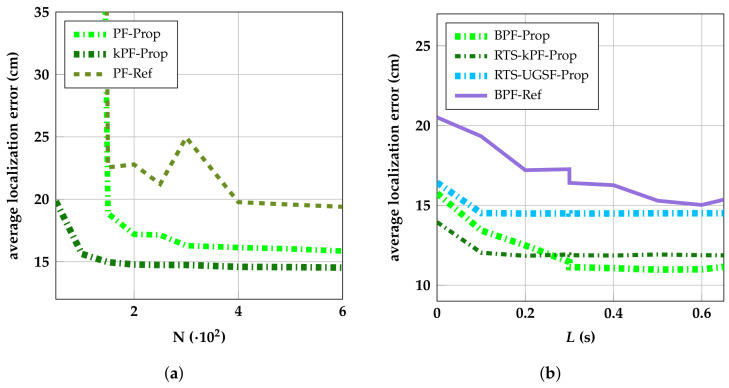
Average position errors of proposed and referenced algorithms as a function of the amount of particles N (**a**) and of the fixed lag *L* in seconds (**b**).

**Figure 10 sensors-23-08289-f010:**
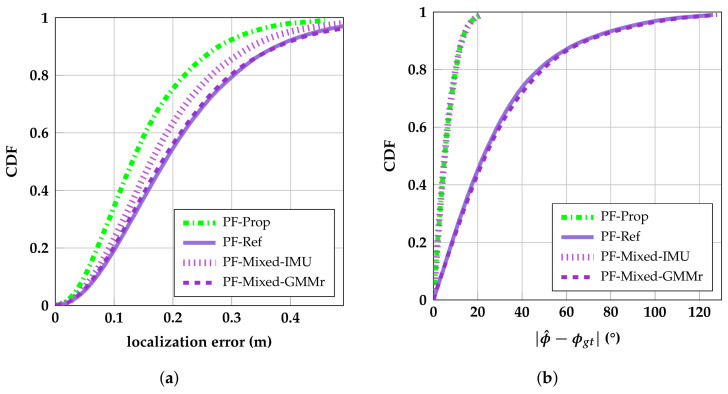
CDFs of the position errors (**a**) and $\hat{\phi }$ errors (**b**) of the proposed, referenced, and mixed PF algorithms.

**Figure 11 sensors-23-08289-f011:**
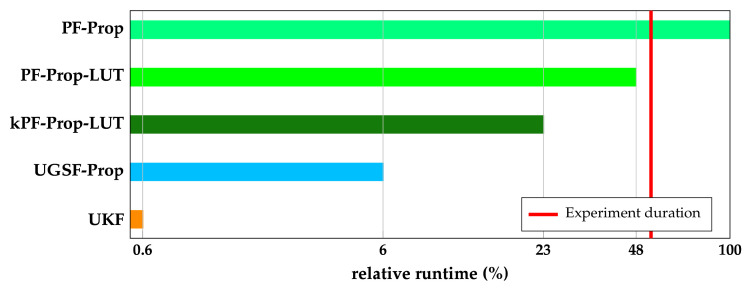
Relative runtime of proposed and benchmark algorithms on a logarithmic scale. One hundred percent equals 98.1 s on a Raspberry Pi (RPi) 4 model B running Python 3.11 for an experiment duration of 53 s.

**Table 1 sensors-23-08289-t001:** UWB hardware and measurement settings.

General	
Environment type	Open space industrial lab
Measurement area	6.4 m × 7.2 m
# Trajectories	2 (performed 5 times each)
Total experiment length	9 min, 812 m
Sensors	UWB, IMU, mocap (ground truth)
UWB	
Hardware	Wi-Pos [42]
# Anchors	8 available, 4 used
Anchor geometry	Rectangular
Tag position	Abdomen
Sample rate (Hz)	23 (8 anchors), 11.5 (4 anchors)
# Measurements	12,482 (8 anchors), 6241 (4 anchors)
UWB Channel	5
Pulse repetition frequency (MHz)	64
Bitrate (kb/s)	850
Preamble length (bytes)	1024
IMU	
Hardware	9-DoF Adafruit BNO055
Sample rate (Hz)	100
Mocap (Ground Truth)	
Hardware	6-DoF Qualisys system (7 cameras)
Sample rate (Hz)	90
Tracking rate (%)	90%

# = amount of.

**Table 2 sensors-23-08289-t002:** Implemented configurations of the proposed, state-of-the-art, and well-known filter algorithms evaluated in Section 4.2.

Algorithm	Originality	Description
PF-Prop	New	PF with proposed robust GMM-based range error model and IMU-based orientation
kPF-Prop	New	PF-Prop with Kalman prediction step
UGSF-Prop	New	UGSF with proposed robust GMM-based range error model and IMU-based orientation
PF-Ref	State-of-the-art	Referenced PF fitting the range error model to our training data [27]
PF-Ref (unfit)	State-of-the-art	Referenced PF [27] using range error model parameters from the paper
PF-Mixed-IMU	Mixed	PF with IMU-based orientation and referenced [27]
PF-Mixed-GMMr	Mixed	PF with proposed range error model and position-based heading estimation of [53] as used in [27]
LLS	Known	Stateless Linearized Least Squares (LLS) multilateration
EKF/UKF	Known	Default extended/unscented Kalman filter with CV process model
UKF-AS	New	UKF with orientation-aware anchor selection by thresholding ϕ
BPF-*	Known	PF-* algorithm, but with a Backtracking PF, thus acting as a smoother
RTS-*	Known	Rauch-Tung-Striebel (RTS) smoother implemented on top of * algorithm

*: wildcard for foregoing algorithm abbreviations

## Data Availability

Data can be shared on request.

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
