# Peer review of "Robust IMU-Based Mitigation of Human Body Shadowing in UWB Indoor Positioning"

_sensors, 2023, doi:10.3390/s23198289_

Round 1
Reviewer 1 Report
Ultra-wideband (UWB) indoor positioning systems can achieve impressive sub-decimeter-level accuracy, but their performance is hindered under Non-Line-of-Sight (NLoS) conditions, especially due to human body shadowing (HBS). This paper introduces an HBS mitigation method that uses the orientation of the body and tag in relation to UWB anchors. By integrating a robust range error model with a tracking algorithm and using a bank of Gaussian Mixture Models (GMMs) based on the body-tag-anchor orientation, the proposed system enhances accuracy. This orientation is determined through an Inertial Measurement Unit (IMU) attached to the tag. Tested in an industrial lab setting, the introduced algorithms notably surpass existing ones, reducing error by 37%.
The research work has addressed an important challenge in the domain of UWB indoor positioning systems, which is the degradation of performance under NLoS conditions, specifically caused by human body shadowing (HBS). The problem is timely and relevant, and the focus on pedestrian localization is a practical application scenario. The paper can be improved on several aspects, as follows:
1. Real-world Application Scenarios: A discussion or case study on potential real-world applications, beyond the industrial lab environment, might make the research more relatable and showcase its broader implications.
2. Comparative Analysis: While the paper mentions that their approach outperforms other state-of-the-art algorithms, it would benefit from a detailed comparative analysis, discussing the specific strengths and weaknesses of each approach.
3. Limitations: The paper might benefit from a clearer discussion on potential limitations or scenarios where the HBS mitigation strategy may not perform optimally. Are there conditions under which the IMU may provide inaccurate orientation data? How does the system handle these scenarios?
4. Scalability: How scalable is the proposed solution? Can it be easily implemented in larger environments or with a larger number of users without significant degradation in performance?
5. Variability of Human Movement: Pedestrian movement can be highly variable. A deeper dive into how the model accounts for diverse and unpredictable human behaviors might strengthen the research.
Minor editing of the language is required.
Reviewer 2 Report
The Authors propose an inertial measurement unit-based human body shadowing mitigation approach for ultra-wideband localization, implemented on top of two known tracking algorithms, i.e. the particle filter and the unscented Gaussian sum filter. They compare their results with those of other commonly used tracking algorithms. The Authors show that both the algorithms they propose outperform all other algorithms, with the particle filter variant achieving the highest accuracy. However, the Authors observe that the unscented Gaussian sum filter variant is considerably faster, thus making it more suitable for constrained, low-power devices for e.g. Internet of Things applications. Furthermore, the paper goes on to analyze how the proposed method performs when combined with smoother algorithms, showing how to further improve their accuracy. The paper is original, rather innovative and the content is quite significant. The presentation is clear and it conveys a good sense that the scientific approach adopted is very sound and the obtained results are robust. The manuscript is likely to attract the interest of a large audience of specialized readers, having a chance to attract a fair numebr of citations. It should be published in its present form.
1. The methodology implemented by the Authors is in line with the serious requirements for a scientific investigation, nence nofurther improvements are needed in this regard.
2. The conclusions are definitely consistent with the evidence and arguments presented by the Authors. Moreover they fully address the main question posed in the manuscript.
3. I find the references to be fully appropriate.
4. The two ables and the eleven figures indeed are fine.
Author Response
Thank you for reviewing our manuscript.
Kind regards,
Cedric
Reviewer 3 Report
Comments to the Author
This paper proposes an human body shadowing mitigation strategy based on the orientation of the body and tag relative to the UWB anchors. It is an interesting research topic with many potential application areas. However, there are several points that need to be addressed to improve the quality of the manuscript.
Suggestions to improve the quality of the paper are provided below:
1. It was mentioned very briefly in the first paragraph of the Introduction section that indoor localisation has many applications, including pedestrian navigation and automatic inventarisation with drones. However, there are many other popular application areas that were not discussed, including emergency management, smart energy management, smart HVAC controls, point-of-interest identification, and occupancy prediction. I suggest that the authors review the following established works as a good starting point to highlight more application areas where indoor positioning systems are leveraged to cater to a wider audience.
Indoor localisation for building emergency management
10.1109/IUCC-CSS.2016.013
Indoor localisation for smart energy management
https://doi.org/10.1016/j.buildenv.2022.109472
Indoor localisation for smart HVAC controls
https://doi.org/10.1145/2517351.2517370
Indoor localisation for point-of-interest identification
https://doi.org/10.3390/ijgi10110779
Indoor localisation for occupancy prediction
https://doi.org/10.1016/j.buildenv.2022.109689
2. While the authors have listed out a few indoor localisation technologies in the Introduction section, it appears to be lacking as they directly go on to highlight the advantages of the UWB-based approaches, without any discussions about the other indoor localisation technologies. Out of this list, Wifi-based and Bluetooth Low Energy-based approaches are identified by [1] to be particularly popular due to their low energy consumption and high detection accuracy, similar to UWB. Please kindly review the following paper as a good starting point and compare between these three technologies (i.e., Wifi, Bluetooth Low Energy, and UWB), before concluding why UWB is selected for this study.
[1] https://doi.org/10.1016/j.buildenv.2020.106681
3. The authors have listed out a few realisations that were achieved as part of this study. However, instead of listing out what is done in this paper, it should be rephrased to more clearly highlight the contribution and novelties of this work.
4. In the last Discussion section, please provide a detailed discussion about the following questions/limitations and how they will be addressed in future works:
· It was mentioned by the authors that future works will extend the evaluation of the proposed algorithm to other environments, multiple people, and more tag positions. However, it is unclear how these variables would have an impact, and the extent of this impact, on the localisation performance of the proposed system.
· It was mentioned in Section 3.1.5 that the tags are held directly in front of the body at the abdomen area, while walking casually, and arms are kept to the side. These seem to be very controlled conditions and not a good reflection of real world deployments where occupants may wear the tags in different ways and walk at different speeds. How will these variables be addressed in future works?
5. Minor comments
· The section headers for Section 3.1.1 and 3.1.5 are not capitalised.

There are no major issues related to the manuscript's quality of English, except for some minor issues highlighted in my current set of comments.
Reviewer 4 Report
1. The introduction of the research gap is insufficient, although the shortcomings of other studies [7-12] are mentioned in the paper but not explained in detail. Moreover, some of the papers are five years old, so it is suggested to use the latest research to prove the gap in this paper. still exists
2. Figure 2(b) shows that the UWB anchors are placed on the wall, which is inconsistent with the text's description.
3. According to the ground truth, there are 3 LOS anchors in the experimental process in most cases, which is already enough for UWB to carry out accurate localization. Can you verify that your proposed algorithm has higher localization accuracy than simply removing the NLOS anchor?
Minor editing of English language required
Round 2
Reviewer 1 Report
The paper has significantly been improved by addressing the reviewers' comments. No further revision is required.
Reviewer 3 Report
Thank you for taking the time to address my comments thoroughly and comprehensively. I believe all my comments have been adequately addressed, and the quality of the manuscript has increased significantly as a result. I have determined that the manuscript is now ready for publication.
There are no major issues related to the manuscript's quality of English, except for some minor issues that do not affect the clarity and flow of the manuscript.